# The Role of Psychobiotics to Ensure Mental Health during the COVID-19 Pandemic—A Current State of Knowledge

**DOI:** 10.3390/ijerph191711022

**Published:** 2022-09-03

**Authors:** Dorota Zielińska, Marcelina Karbowiak, Aneta Brzezicka

**Affiliations:** 1Department of Food Gastronomy and Food Hygiene, Institute of Human Nutrition Sciences, Warsaw University of Life Sciences (WULS-SGGW), Nowoursynowska 159C Str., (Building No. 32), 02-776 Warsaw, Poland; 2Neurocognitive Research Center, SWPS University of Social Sciences and Humanities, Chodakowska Str. 19/31, 03-815 Warsaw, Poland

**Keywords:** COVID-19 pandemic, mental health, probiotics, psychobiotics

## Abstract

Psychobiotics are defined as probiotics, mainly of the genus *Lactobacillus* and *Bifidobacterium*, that confer mental health benefits to the host when consumed in a particular quantity through the interaction with commensal gut microbiota. The gut microbiota, which means a diverse and dynamic population of microorganisms harboring the gastrointestinal tract, communicates with the brain and vice versa through the brain–gut axis. The mechanisms of action of psychobiotics may be divided into four groups: synthesis of neurotransmitters and neurochemicals, regulation of the HPA axis, influence on the immune system, and synthesis of metabolites. Recent years showed that the COVID-19 pandemic affected not only physical, but also mental health. Social isolation, fear of infection, the lack of adequate vaccine, disinformation, increased number of deaths, financial loss, quarantine, and lockdown are all factors can cause psychiatric problems. The aim of this review was to discuss the potential role of psychobiotic in light of the current problems, based on *in vitro* and *in vivo* studies, meta-analyses, clinical trials evidence, and registered studies assessing probiotics’ therapeutic administration in the prevention or treatment of symptoms or side effects of COVID-19.

## 1. Introduction

The COVID-19 pandemic has contributed to the global rise in chronic diseases and associated risk factors, including obesity, high blood sugar, and outdoor air pollution, which caused the storm in the last 30 years resulting in the deaths of many people. Before 2020, mental disorders were one of the leading causes of the global burden of health, with depression and anxiety disorders among the 25 leading causes of burden worldwide. The outbreak of the COVID-19 pandemic has created an environment in which many of the determinants of mental health are exacerbating. Up-to-date information is needed on potential measures to improve mental health, especially regarding the effects of COVID-19 [1]. This review aimed to discuss the potential role of psychobiotics in light of the current problems, based on *in vitro* and *in vivo* studies, meta-analyses, clinical trials evidence, and registered studies assessing probiotics’ therapeutic administration in the prevention or treatment of symptoms or side effects of COVID-19.

## 2. Psychobiotics and Their Mechanisms of Action

### 2.1. Probiotics vs. Psychobiotic

Recent years have provided a lot of evidence on the microbiota effects on human health. The human gut microbiota is a complex of bacteria, viruses, protozoa, archaea, and fungi that inhabit the gastrointestinal tract [2]. In this context, the role of probiotics in modulating human gut microbiota is arising. Probiotics are live organisms, which when administered in adequate amounts offer health benefits to the host [3]. A strain that can be considered a probiotic ought to demonstrate the safety of the host organism and the ability to colonize the digestive system. The basic criteria in assessing whether a given strain can be classified as probiotic are also such factors as the survival of bacteria under low pH conditions, in the presence of gastric acid and proteolytic enzymes. Moreover, widespread mechanisms of probiotic action can be similar across taxonomic groups, such as inhibition of potential pathogens or the production of useful metabolites or enzymes. Other effects, including neurological, immunological, and endocrinological properties, are more likely to be strain-specific and are rather rare [3].

Currently, much attention is paid to the influence of probiotics on the nervous system. The latest evidence suggests that gut microbiota not only regulates intestinal function and health but also plays a role in mental health through the gut–brain axis [4]. In 2013 Professor Ted Dinan of University College Cork, Ireland proposed the term ‘Psychobiotics’ that summarized the psychological potential of the molecules produced by probiotic bacteria and refers specifically to probiotics, but also prebiotics and all microbiota-targeted interventions that can manipulate microbiota–gut–brain signals and have positive effects on neurological functions such as mood, cognition, and anxiety [5]. Psychobiotics correspond to a specific class of probiotics, mainly of the genus *Lactobacillus* and *Bifidobacterium*, capable of producing neuroactive substances such as γ-aminobutyric acid (GABA) and serotonin, which exert effects on the brain–gut axis [6]. Many probiotic microorganisms have been proposed as potential psychotropic agents, including *Streptococcus thermophilus, Bifidobacterium animalis*, *Bifidobacterium bifidum*, *Bifidobacterium longum*, *Streptococcus thermophiles*, *Lactobacillus bulgaricus*, *Lactococcus lactis*, *Lactobacillus acidophilus*, *Lactobacillus plantarum*, *Lactobacillus reuteri*, *Lactobacillus paracasei*, *Lactobacillus helveticus*, *Lactobacillus rhamnosus*, *Bacillus coagulans*, *Clostridium butyricum*, and others [7,8,9,10]. 

### 2.2. The Gut–Brain Axis

Over 2000 years ago the Greek physician-philosopher Hippocrates said: “All disease begins in the gut”. During the past years, researchers discovered the unique relationship between gut microbiota and the brain and called it the gut–brain axis. The first mention of this came from the case of an army surgeon when the relationship between intestinal and mood has been found [4]. The mechanism by which the bacteria inhabiting the gut act on the brain is not fully understood. There are a few hypothetical mechanisms; some of them have already been well ascertained, while the majority need more detailed analysis. 

The gut–brain axis is based on bidirectional interaction between gut microbiota and the brain. Gut microbiota communicates with the brain and vice versa through the central, autonomic, and enteric nervous systems, as well as the hypothalamic–pituitary–adrenal (HPA) axis [11]. Efficient communication is possible not only due to the nervous system, but also through cytokine released by mucosal immune cells, hormones produced by endocrine cells, or through the vagus nerve. On the other hand, signals related to stress such as stress hormones or sympathetic neurotransmitters and neurochemicals influence the gut microbiota [12]. Several mechanisms may be responsible for the potential psychobiotic effects elicited by the gut microbiota: synthesis of neurotransmitters and neurochemicals, regulation of the HPA axis, influence on the immune system, and synthesis of metabolites (Figure 1). 

#### 2.2.1. Neurotransmitter and Neurochemical Substances 

Microbiota may secrete many kinds of neurotransmitters, such as acetylcholine, serotonin (5-HT), dopamine, noradrenaline, GABA, glycine, catecholamine, and other substances [13]. Neuroactive molecules and signal molecules secreted by intestinal microbiota affect the amplification of the signal on the gut–brain axis. When the synthesis of intestinal neurotransmitters increases the plasma tryptophan decreases, which in turn causes an increase in the transmission of molecules to the brain and improvement in patients suffering from mental and neurodegenerative diseases. Moreover, neuroactive molecules secreted by intestinal microbiota can regulate nerve signals and may directly or indirectly modulate neuropsychiatric parameters such as sleep, appetite, mood, and cognition [14].

Serotonin is recognized as one of the most important neurotransmitters that can be secreted by intestinal bacteria, mostly by probiotic *Lactobacillus* and *Bifidobacterium*: 5-HT is synthesized from the essential amino acid tryptophan in the serotonergic neurons of the central nervous system (CNS) and the enterochromaffin cells of the gut. Different research points to the contribution of the intestinal microbiota in the regulation of the levels of 5-HT in the intestine and the gut–brain axis. More than 90% of all 5-HT in the body is synthesized in the intestine, where it plays a potential regulatory role in the gut by activation receptors located in enterocytes, enteric neurons, and cells of the immune system [15]. An important role in the pathophysiology of depression stems from the 5-HT involving hippocampal brain-derived neurotrophic factor (BDNF) expression signaling mechanism. BDNF is a growth factor crucial in brain plasticity, learning and memory functions, and is abnormally reduced in patients suffering from depression [16]. It was found that 5-HT stimulates BDNF expression while BDNF promotes the neurogenesis and neuronal survival of 5-HT; impairment in this mechanism is claimed to cause depression symptoms [17]. The administration of psychobiotic may upregulate the 5-HT-BDNF system to mediate the anti-depressive effect via different neural and immune-mediated humoral pathways [18]. Moreover, it has been shown that probiotics have a positive effect on the tightness of the intestinal barrier, reduce the stress response, and increase the level of 5-HT and the expression of GABA receptors in the CNS [19,20,21]. The administration of probiotics increases the expression of GABA receptors in mice in the areas of the brain responsible for emotions, reducing stress-induced anxiety and depression behavior in rodents [7,9,22]. For example, it was found that the administration of GABA-producing *Lactobacillus brevis* strain significantly improved insomnia and had anti-depressive effects, comparable to fluoxetine in mice models [23,24]. *Lactobacillus* and *Bifidobacterium* strains are found to produce large quantities of GABA in the presence of suitable substrates [25]. What is more, probiotics can convert glutamate to GABA, which is an inhibitory neurotransmitter, and may inhibit the transmission of pain messages from the CNS [26].

In terms of other neurotransmitters or neurochemical substances, there are only a few shreds of evidence that proved the psychobiotics’ regulatory role, for example, noradrenaline level by *Bifidobacterium infantis* strain [27]; dopamine level by *L. plantarum* strain [28]; norepinephrine level by *L. helveticu, L. casei, L burgaricus* [29]; acetylcholine level by *L. plantarum* strain [30]; or histamine level by *L. plantarum* and *L. reuteri* strains [31]; and glutamate level by many *Lactobacillus* strains [32]. Moreover, a few probiotic strains can utilize nitrate to synthesize nitric oxide (NO), which regulates the responses to different levels of the immune and nervous systems. Some of the psychobiotics also enhance the activity of the enzyme indoleamine-2,3-dioxygenase (IDO), which is employed in the catabolism of tryptophan and formation of kynurenic and quinolinic acid—the neuroactive compounds that are a means to affect the neuronal pathways to the brain and correlate positively with the symptoms of depression [33].

#### 2.2.2. The Influence on the Immune System

The digestive tract constitutes approximately 70% of the immune system [34]. When the gut microbiome is in a state of homeostasis, the body can fight inflammation more effectively by stimulating the production of anti-inflammatory cytokines. This means that continued exposure to increased levels of neurotransmitters, as well as imbalances in the intestinal tract and changes in permeability, leads to the production of a kind of proinflammatory endotoxin—lipopolysaccharide (LPS), which may exacerbate mental disease [35]. LPS regulates emotion by increasing the activity of the amygdala, affecting the physical activity of the brain, and regulating the production of neuropeptides. Even small amounts of LPS can stimulate proinflammatory cytokine secretion and increase IDO activity and norepinephrine in plasma, which relates to depression symptoms [36]. 

It should be emphasized that there is a relationship between the development of depression, the immune response, and the function of the gut [21,37]; this is called a leaky gut syndrome. Research has revealed that stress impairs the tight junctions (connections between intestinal epithelial cells), which in turn leads to the translocation of intestinal bacteria through the intestinal barrier into the circulatory system. LPS is a component of the outer membrane of Gram-negative bacteria and can penetrate and even damage the blood–brain barrier, consequently allowing other substances, like toxins or pathogens to cross the barrier, causing inflammation [38]. In contrast, the presence of probiotic strains of bacteria, as well as a mucin degrading bacteria—*Akkermansia muciniphila*—may improve intestinal barrier function by increasing the expression of a protein involved in the tight junction [39].

Stress and depression are related to high levels of proinflammatory biomarkers like interleukin (IL): IL-1β, IL-1α, IL-6, tumor necrosis factor-α (TNF-α), interferon-γ and C-reactive protein (CRP) [40]. In many studies, it has been shown that probiotics can effectively raise anti-inflammatory cytokines and reduce pro-inflammatory cytokines by regulating signaling pathways such as nuclear energy source-kappaB (NFκB) and mitogen-activated protein kinase (MAPK) [41]. Moreover, the onset of a clinical report indicated a correlation between probiotic intake and the activation of macrophages, NK cells, antigen-specific cytotoxic T-lymphocytes, and the release of various cytokines in humans and more specifically in elderly [42] and infant subjects [43]. Probiotics may also regulate the immune system by acting on T-regulating cells and the secretion of IL-10, which is important in immune regulation activity [44]. For example, *Lactobacillus rhamnosus* GG (LGG) has *in vitro* effects on enhanced IL-10 release of mononuclear cells [45]. Also, the administration of LGG was shown to be able to increase IL-10 in some patients [46].

Fatigue is a symptom of stress and is also related to mental health issues. A higher concentration of reactive alternative form (ROS) and consequently oxidative stress play important role in the pathogenesis of fatigue [47]. Psychobiotics may: reduce inflammation and oxidative damage due to their ability to chelate metal ions, antioxidant systems such as activating superoxide dismutase and catalase; regulate signaling pathways, including MAPK; and lower ROS scavenging proteins [48].

#### 2.2.3. Regulation of the HPA Axis

The HPA axis plays a crucial role in the two-way communication of the gut–brain axis. This axis is considered to be the spinal efferent stress axis that coordinates the body’s adaptive responses to all kinds of stressors and is also part of the limbic system, a key area of the brain primarily involved in memory and emotional responses [49]. Stress can lead to changes in the microbiota and consequently mood through the HPA axis. Chronic stress activates the HPA axis, which ultimately stimulates the release of glucocorticoids from the adrenal cortex in response to stimulation by an adrenocorticotropic hormone from the anterior pituitary gland. When glucocorticoid levels increase, then the inhibition of the secretion of corticotropin-releasing hormone and vasopressin in the hypothalamus act creating a negative feedback loop. Furthermore, stress-induced intestinal dysbiosis exacerbates inflammation and intestinal permeability and stimulates the release of pro-inflammatory cytokines which activate the HPA axis [18]. 

Psychobiotics can play both direct and indirect roles in weakening or normalizing HPA axis hyperactivity by restoring intestinal dysbiosis and modulation at the neurotransmitter level. When the HPA axis is dysfunctional, the production and function of stress-related hormones are disrupted [19,20]. Probiotics can act as an antidepressant by downregulating the HPA axis or increasing the production of tryptophan as a serotonin precursor [37,50]. For example, a probiotic combination of *L. helveticus* and *Bifidobacterium longum* significantly improved depressive-like behaviors in mice models by reducing corticosterone, as well as cortisol levels, which have been linked to the attenuation of HPA-axis hyperactivity. Moreover, Bravo et al. [51] found that *Lactobacillus rhamnosus* reduced the increase in cortisone induced by stress. Moreover, it was observed that some psychobiotics prevented the stress-induced reduction in hippocampal neurogenesis of noradrenaline and restored the gut barrier [52]. 

#### 2.2.4. The Other Metabolite Synthesis

Another possibility of the action of psychobiotics on the gut–brain axis may be short-chain fatty acids (SCFAs), which are formed by the fermentation of soluble fibers such as galactic- and fructo- oligosaccharides. These SCFAs include acetate, propionate, and butyrate and are produced by *Lactobacillus, Bifidobacterium, Propionibacterium, Eubacterium, Bacteroides, Clostridium, Roseburia,* and *Prevotella* species [53]. SCFAs have a range of regulatory activities beneficial to the host, notably the regulation of systemic energy homeostasis and colonocyte metabolism [54], immune regulation, mitochondrial biogenesis, and the maturation and function of the brain’s microglia, the brain’s most abundant immune cells associated with brain inflammation. SCFAs also act as epigenetic modulators through histone deacetylases, contribute to the anti-inflammatory and antioxidant effects of probiotics and markedly reduce the markers of inflammation [55]. Moreover, it was recently found that the SCFA plays a crucial role in microglia maturation, morphology, and immunological functions, which is important due to microglia being the primary innate immune effector cells of the CNS [56].

Butyrate serves as an inhibitor of histone deacetylase (HDAC), resulting in the loosening of the chromatin and increasing the acetylation process. Therefore, epigenetic signatures are altered, facilitating the access of DNA repair enzymes to the transcription of various regulatory genes [57]. The butyrate-producing probiotics *L. plantarum*, *B. infantis*, *Clostridium butyricum,* and *F. prausnitzii* have been shown to ameliorate depressive behaviors in mice models, while increased BDNF expression was observed in experiments with *B. infantis* and *C**. butyricum* [58,59].

Histamine is a biogenic monoamine that is produced by Enterochromaffin (EC) cells in the gastrointestinal tract (GIT); it has been recently found to be a product of gut microbial metabolism especially certain *Lactobacillus*, *Lactococcus*, *Streptococcus*, *Pediococcus*, and *Enterococcus* spp. It plays a role in a wide variety of physiological functions including cell proliferation, allergic reactions, wound healing, and regulation of immune cells, as well as acting as a neurotransmitter in the brain [60]. Histamine can have either pro- or anti-inflammatory properties, which relate to the receptor that it acts upon. In the brain, overproduction of histamine induces an allergic inflammatory response by increasing the production of proinflammatory cytokines such as IL-1α, IL-1β, IL-6, and various chemokines. On the other hand, in the rats with vascular dementia model, a deficit of histaminergic signaling was found [61].

## 3. The Influence of the COVID-19 Pandemic on Mental Health 

### 3.1. Causes and Effects of Mental Health Disorders 

Since December 2019 COVID-19 has become an international public health emergency caused by severe acute respiratory syndrome coronavirus 2 (SARS-CoV-2). The typical clinical symptoms of the diseases are fever, cough, and myalgia/fatigue, as well as gastrointestinal symptoms, such as diarrhea. Complications may develop the disease into acute respiratory distress syndrome (ARDS) and result in death [62]. In addition, the COVID-19 outbreak can also affect mental health causing problems such as stress, anxiety, depression, sleep disorders, lower mental well-being, panic, posttraumatic stress disorder (PTSD), and even suicide [63,64,65].

The main stressful factors that can be described as causes of mental health problems are social isolation (extended lockdown, quarantine), the increase in the number of deaths, or deaths of family and friends, fears of infection, the lack of a possible vaccine for COVID-19, uncertainties of the future, the job losses, disinformation, and the overall negative environment [66]. Social factors, including the quality of government intervention, are also major predictors of mental health problems as a reaction to the COVID-19 pandemic. The prevalence of depressive symptoms was significantly lower in countries where governments promptly implemented stringent policies [67]. 

The WHO claims a 25% increase in mental health problems, especially depression and anxiety, after the COVID-19 pandemic outbreak. Some studies indeed reported an increase in the vast range of mental health issues and pointed to loneliness as the main cause [68]. What is important, it did not affect all populations equally: women and adolescents seem to be the most affected groups [69,70]. What is interesting is that several meta-analyses and systematic reviews do not confirm the strong statement about the massive outbreak of mental health issues in reaction to the pandemic [71]. For example, in a review article, Robinson et al. [71] conclude that: “There was a small increase in mental health symptoms soon after the outbreak of the COVID-19 pandemic that decreased and was comparable to pre-pandemic levels by mid-2020 among most population sub-groups and symptom types”. Another meta-analysis points out that it was a rather small increase in mental health symptoms mostly visible soon after the outbreak of the COVID-19 pandemic. What is even more important is that the levels of depression and anxiety scores were comparable to those of pre-pandemics by mid-2020 in all researched populations, but people with physical health conditions seemed more prone to mental health issues during the pandemic [71]. However, it should be underlined that extended social distancing influenced mental health significantly by decreasing social bonding and physical expressions. There was also the cause of the biochemical pathways’ imbalance of neuropeptides (e.g., oxytocin), endocannabinoids (e.g., anandamide), and corticosteroids (e.g., cortisol), which are the key substances involved in the regulation of the immune response and in maintaining emotional wellbeing [72].

The effects of the COVID-19 pandemic can be divided into simply reactions to the pandemic and restriction measures [73]. The main effects concerning mental health are loneliness, anger, anxiety, depression, and panic. Moreover, another mental problem, which is more of a cognitive nature, is the so-called “brain fog” reported in COVID-19 patients even after experiencing a mild infection [74]. This state comprises symptoms of impaired attention, memory, speed of information processing, and dis-executive function, and is similar to the condition known as a “chemo fog”—a mental state reported in cancer patients undergoing chemotherapy [74]. Those similarities led Michelle Monje and Akiko Iwasaki, of Stanford and Yale Universities, respectively, to the idea of looking into the immune-neural mechanism common for cancer and COVID-19 survivors. Their teams showed that in mice with mild COVID-19 infections, the virus disrupted the normal activity of several brain cell populations and left behind signs of inflammation [74]. In this study, mice experienced mild respiratory COVID and showed persistently impaired hippocampal neurogenesis, decreased oligodendrocytes, and myelin loss together with elevated cytokines (including CCL11). The authors of this study also showed that systemic CCL11 administration specifically caused hippocampal microglial reactivity and impaired neurogenesis. What is more, people with cognitive problems after COVID-19 infection show elevated CCL11 levels. This study showed a link between inflammatory response and neural degeneration that may be a cause of observed mental health issues among COVID-19 survivors.

### 3.2. Gut Microbiota Dysbiosis

The link between different lung diseases and dysbiosis has been reported previously and is called the gut–lung axis [75,76]; however, gut microbiota changes in patients suffering from COVID-19 were recently shown [77,78]. The first evidence that there may be an association between gut bacteria imbalance and COVID-19 severity was presented by Gu et al. [77]. The study of patients with COVID-19, influenza A (H1N1), and healthy volunteers was conducted to identify differences in the gut microbiota by 16S ribosomal RNA gene V3-V4 region sequencing. Compared with the control group, COVID-19 individuals exhibited a significantly higher relative affluence of bacterial opportunistic pathogens and a lower relative abundance of beneficial microorganisms [77]. In another study, carried out in the first months of the COVID-19 pandemic, significant alterations in GI microbiota of COVID-19 patients, persistent dysbiosis despite cure, and a positive correlation between dysbiosis and severity of COVID-19 were found [79]. A similar result was obtained also by Yeoh et al. [80], who demonstrated that gut microbiome composition was significantly altered in patients with COVID-19 compared with healthy individuals. Several gut microorganisms such as bifidobacteria, with well-known immuno-modulatory potential, were underrepresented in COVID-19 patients and remained low in samples collected up to 30 days after cure. Authors suggested that it certainly was involved with elevated concentrations of inter alia inflammatory cytokines [80]. A correlation between the relative abundance of some indicator microorganisms such as *Coprobacillus, Clostridium ramosum*, and *Clostridium hathewayi,* and the severity of COVID-19 has been found. Furthermore, an inverse relationship between the abundance of *Faecalibacterium prausnitzii* and the disease severity was denoted [81]. *F. prausnitzii* is a commensal bacterium that plays a role in intestinal homeostasis by inhibiting the production of proinflammatory cytokines and can stimulate high secretion levels of IL-10 through peripheral blood mononuclear cells, mucosal dendritic cells, and macrophages. It is also suggested that *F. prausnitzii* may contribute to host immunology defense in patients with COVID-19 [82].

Zhao et al. [83] noticed that several studies reported that infected SARS-CoV-2 virus patients suffer from gastrointestinal symptoms and intestinal dysbiosis may affect the failure to respond to vaccines, which led them to summarize the accumulated knowledge concerning the function of the gut microbiota in the host’s immune response [83]. The results of those studies highlighted the close relationship between the microbiome composition and the SARS-CoV-2 infection. The studies exhibited evidence of substantiating the modulation of the intestinal microbiota as supportive therapy for COVID-19. Though continually limited in several findings in which the inference might be biased, studies to date have consistently demonstrated gut dysbiosis in patients with COVID-19. Although there is no direct clinical evidence yet showing the efficacy of any specific strain of probiotic against COVID-19, several ongoing registered trials study the role of gut microbiota in COVID-19 or the therapeutic potential of various probiotic formulations in COVID-19. Currently (July 2022), all ongoing clinical trials registered at ClinicalTrials.gov investigating the role of probiotics in the prevention, treatment, recovery, and side effects from COVID-19 disease or before or after vaccination against COVID-19 have been collected and described in Appendix A.

### 3.3. Current Treatment Strategies

As mentioned above, the COVID-19 pandemic leads to psychiatric problems regarding the occurrence of various stressful factors, and post-COVID memory, attention, and executive function problems. Various meta-analyses of COVID-19 patients, i.e., people affected by the COVID-19 pandemic and services involved in fighting the pandemic, have pointed to the fact that psychological ailments like anxiety, depression, and PTSD affect them and this poses a challenge for the healthcare system [65,84]. In this light, there is an apparent necessity for the urgent development of effective methods to take care of the well-being of people facing mental health problems caused by the SARS-CoV-2 virus. It seems, therefore, that it is impossible to avoid the use of psychotropic medications in a population predisposed to the occurrence of mental disorders. The latest findings indicate an increase in new prescriptions for antidepressants and anxiolytics for individuals during COVID-19 mainly in females and people aged 40 years and older, with the highest rates of use in the population 80 years and older [85]. Nevertheless, endocrine and metabolic effects are associated with psychotropic medication (antipsychotics, antidepressants, and mood stabilizers), including hyperprolactinemia, hyponatremia, diabetes insipidus, hypothyroidism, hyperparathyroidism, sexual dysfunction and virilization, weight loss, weight gain, metabolic syndrome, and changes in the composition of the gut microbiota and gastrointestinal function [86]. In addition, the latest studies also indicate the adverse events of psychotropic medications influencing the weakening of the immune system or the risk of developing dementia. Nemani et al. [87] demonstrated that there is a link between the use of psychotropic medications and the increased risk of COVID-19 infection [87]. Concurrently, pre-COVID-19 psychotropic medication exposure was associated with a greater 1-year incidence of dementia after COVID-19 infection [88]. For this reason, the possibilities of treating mental disorders with “green” or “natural” alternatives with an acceptable safety profile have been sought for a long time.

## 4. Can Psychobiotics Minimize the Mental Health Disorders Connected with the COVID-19 Pandemic? What Do We Know?

A search of ongoing clinical trials at the ClinicalTrials.gov website, maintained by the US National Library of Medicine (NLM) at the National Institutes of Health (NIH), reveals as many as five registered studies assessing probiotics therapeutic administration in the prevention or treatment of symptoms or side effects of COVID-19 with mental health issues outcome measures included (Table 1).

As stated above, the COVID-19 pandemic started unexpectedly, and its after-effects are still being discovered, this means that there are no finished and published clinical data regarding COVID-19 and mental health. Ongoing clinical trials can bring much more information and may accelerate therapeutic psychobiotic strategies. While efforts to formulate direct anti-viral agents and vaccines are of prime significance to cut the risk of infection by the SARS-CoV-2 virus, it must not be forgotten that supporting the gut–brain axis may still have therapeutic potential.

Registered, ongoing clinical trials are based on data from other *in vitro* and *in vivo* studies, which served as the rationale for this research. In the last decade, several experiments have been carried out in animal models and among humans, in which probiotic strains, exerting the potential psychobiotic effects, were used in different neurological issues [20,89,90,91,92,93,94,95]. Several studies have suggested that the administration of probiotics can be effective in stress management. Kato-Kataoka et al. [89] investigated the effects of the probiotic *Lactobacillus casei* strain Shirota (LcS) on psychological, physiological, and physical stress responses in medical students exposed to stress during exams. Inter alia, the day before the exam, salivary cortisol and plasma L-tryptophan levels were significantly increased in only the placebo group (*p* < 0.05), and two weeks after the examination the LcS group had significantly higher fecal serotonin levels (*p* < 0.05) than the placebo group. The results suggest that the daily consumption of fermented milk containing LcS may exert beneficial effects in preventing the onset of physical symptoms in healthy subjects exposed to stressful situations [89]. Similar results were achieved by Allen et al. [20] indicating that consumption of *B. longum* 1714 is associated with reduced stress and improved memory at the same time [20]. Therefore, probiotics seem to play a beneficial role in stressed populations. In support of this, a systematic review and meta-analysis to assess the effects of probiotics on stress in healthy subjects were conducted [96].

Other clinical insights have illustrated that probiotic supplementation may act as an adjuvant strategy to ameliorate or prevent depression. The Akkasheh et al. [90] study was designed to determine the effects of probiotic intake on symptoms of depression and metabolic status in patients with Major Depressive Disorder. A significant reduction in Beck Depression Inventory total scores was observed after eight weeks of intervention for patients who received probiotic supplements compared with the placebo (*p* = 0.001) [90]. In the study by Steenbergen et al. [91], healthy participants without current mood disorders were compared to the placebo group intervention. Administration of the 4-week multispecies probiotics intervention (*Bifidobacterium bifidum* W23, *Bifidobacterium lactis* W52, *Lactobacillus acidophilus* W37, *Lactobacillus brevis* W63, *Lactobacillus casei* W56, *Lactobacillus salivarius* W24, and *Lactococcus lactis* W19 and W58) resulted in a significant reduction in overall cognitive reactivity to sad mood, which was largely accounted for by reduced rumination and aggressive thoughts [91]. To confirm the hypothesis that probiotics administration is associated with a significant reduction in depression, several systematic reviews and meta-analyses of the existing evidence were conducted [97,98,99,100,101,102,103,104,105,106,107,108]. The evidence for probiotics alleviating depressive symptoms is compelling, which is confirmed by the secondary evidence found, but additional double-blind randomized control trials in clinical populations are necessitated to further assess efficacy. The feature of the development of depression is the serotonin hypothesis, which postulates that a reduction in serotonin leads to an increased predisposition to depression [109]. Psychobiotics in such a system represent the supportive therapy enhancing neurotransmitter production in the gut, including serotonin and others such as dopamine, noradrenaline, and GABA, which likely modulate neurotransmission in the proximal synapses of the enteric nervous system [110]. It should be noted that a recent comprehensive and systematic umbrella review of research including systematic reviews, meta-analyses, and large data-set analyses of the evidence on whether depression is caused by lowered serotonin activity or concentrations, revealed that serotonin research provides no convincing evidence of there being an association between serotonin and depression, and no support for the hypothesis that depression is caused by lowered serotonin activity or concentrations [111]. Depression is a heterogeneous disorder with potentially multiple underlying causes with manifold, not single, chemical imbalance in the brain [112]. Psychobiotics capable of producing a wide variety of neurotransmitters may gain more and more importance for the supportive therapy of brain disorders including depression, influencing the various pathways underlying the management of depression. Certainly, future clinical approaches are needed to verify psychobiotics effectiveness in improving mood disorders resulting from the evolution of the COVID-19 outbreak.

There are many severe symptoms associated with depression, including alterations in sleeping patterns. Moreover, there is increasing recognition of inadequate sleep (duration or quality) as a global public health issue, especially intensified during the COVID-19 pandemic [113]. It has been hypothesized that probiotics, prebiotics, and postbiotics might be able to influence sleeping patterns according to recent studies [92,93]. In other studies, lower sleep quality and quantity have been linked to dysbiotic gut microbiomes, which are altered in composition and function [114,115]. It is characterized by a decrease in microbial diversity, an increase in pathogenic microbes, or a loss of beneficial microbes. Sleep-promoting effects of probiotics are mediated by a variety of pathways, including neuronal, hormonal, immune, and clock gene responses. The major role in each of these pathways play butyrate, propionate, and acetate—microbially produced SCFAs [116]. Overall, results of systematic reviews and meta-analyses suggest that probiotics administration can amplify positive modifications to self-perceived sleep health, measured using the Pittsburgh Sleep Quality Index (PSQI), especially in healthy subjects or subjects with mild sleep problems. However, reviews also exhibit that probiotics consumption has no discernible impact on answers to other subjective sleep scales, nor does it affect sleep latency or efficiency [117,118,119]. Nevertheless, it was noted that the available data in this field remains limited. Therefore more research is required, specifically using objective sleep outcome indicators in well-planned populations [119].

Recently some of the research findings presented in preclinical (animal) studies suggest that the ingestion of live microorganisms in sufficient amounts can alleviate the suffering from anxiety [94]. Notwithstanding this fact, the evidence to date for the efficacy of psychobiotics in alleviating anxiety, as presented in currently published randomized clinical trials, is insufficient, perhaps due to the dearth of extant research. In studied meta-analyses of randomized controlled trials, it is pointed out that more reliable evidence from clinical trials is needed before a case can be made for promoting the use of probiotics for alleviating anxiety [120,121,122,123,124].

Lastly, probiotic supplementation was found to improve cognitive function and mood. In a study by Kim et al. [95], psychobiotics containing *Bifidobacterium bifidum* BGN4 and *Bifidobacterium longum* BORI administrated for 12 weeks in a healthy elder population, resulted in greater improvement in mental flexibility test and stress scores than the placebo group (*p* < 0.05) [95]. The effect of the psychobiotic intervention on cognitive function or attenuating cognitive decline was confirmed by meta-analyses of experimental studies [125,126]. The results of systematic reviews also consistently exhibited positive treatment effects [127,128]. Nevertheless, further research is warranted.

A multitude of primary studies showing the effect of probiotics administration on neurological conditions, and secondary evidence summarizing the current state of knowledge (enabling more reliable conclusions to be drawn from the collection of research), as well as the latest reports providing the number of individuals experiencing mental health problems (which additionally escalated during the COVID-19 pandemic [129]), point to the urgent need to find effective ways to address the mental health crisis. It, therefore, seemed rational that when the COVID-19 pandemic began to wreak havoc on the lives of many people, affecting their psychological well-being, they began to consider psychobiotics as a remedy that positively influenced the efficiency of the gastrointestinal tract, but also alleviated symptoms of depression, chronic stress, and improved brain functions. Additionally, due to insufficient data referring to COVID-19 coronavirus disease and the relative safety of probiotics, the prevalence of application of psychobiotics seems reasonable while awaiting further evidence.

The primary benefits of psychobiotics in supporting mental health during crises are that they do not cause the side effects typical of psychotropic drugs, which is especially important in the context of coronavirus disease that affects health even with mild or asymptomatic disease severity. Moreover, psychobiotics are microorganisms that belong to microbial genera naturally found in the gastrointestinal tract, so the risk of allergies and reliance on other treatments is reduced. Administration of strains with psychobiotic characteristics could constitute an alternative to ensure mental health in light of the COVID-19 outbreak, but future clinical trials are necessary to determine their efficacy in treating mental health disorders caused by the influence of the COVID-19 pandemic.

## 5. Conclusions

When the COVID-19 pandemic began in 2019 many stressful life situations hit people around the world. In addition to the symptoms of the virus disease itself, more frequent mental health problems related to protracted restrictions and an unstable world situation have been noted. Previous *in vitro* and *in vivo* studies suggested a link between human gut microbiota and mental health, which was named the gut–brain axis. Moreover, specific strains of probiotic bacteria—psychobiotics, due to their beneficial role, can serve as a “green” alternative for maintaining human mental health. This review summarized the known mechanisms of action, as well as discussed the potential role of selected psychobiotics in mental health disorders. At present, there are only five ongoing clinical studies regarding probiotics administration, COVID-19, and mental health. The scientific community is eagerly awaiting the completion of this ongoing research into the effects of probiotics on various aspects of mental health impacted by the COVID-19 pandemic. In these challenging times, mental health and resilience, which mean the ability to maintain homeostasis, are crucial but frequently neglected components of well-being. However, it needs to be borne in mind that is not possible to provide an evidence-based recommendation for psychobiotics use to prevent or treat mental health illnesses in the COVID-19 pandemic, until clinical trials investigating the prophylactic and therapeutic potential of psychobiotics are complete.

## Figures and Tables

**Figure 1 ijerph-19-11022-f001:**
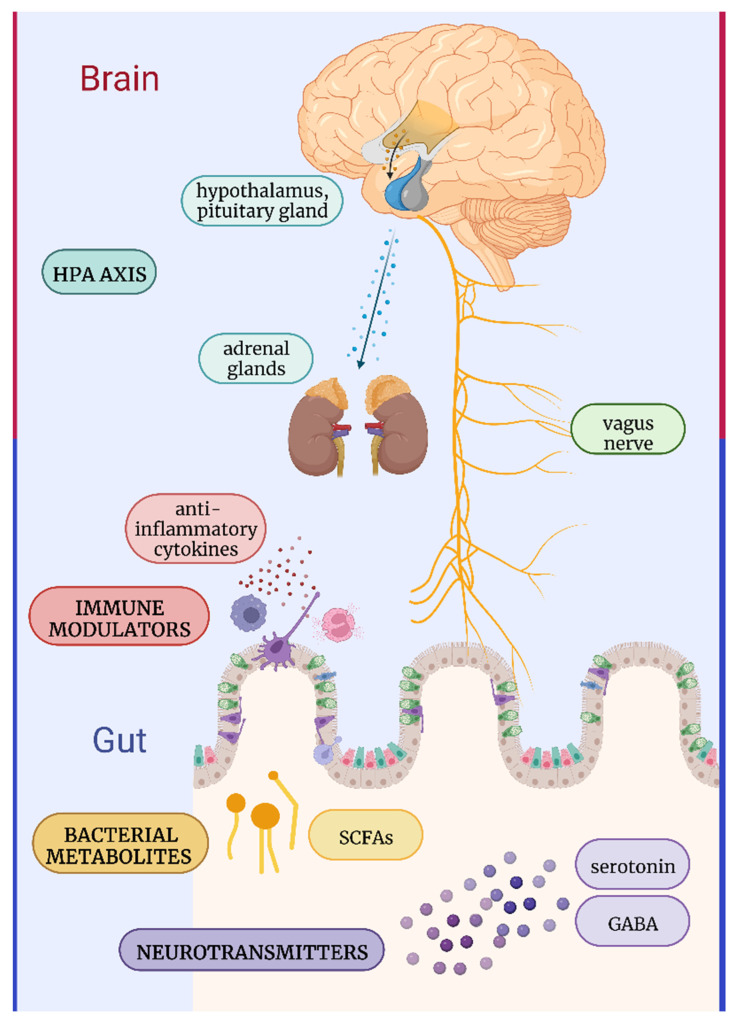
The mechanisms of action by which the gut microbiota elicit the potential psychobiotic effects—regulation of the HPA axis, influence on the immune system, synthesis of neurotransmitters and neurochemicals, and synthesis of metabolites. HPA axis—the hypothalamic–pituitary–adrenal axis; SCFAs—short chain fatty acids; GABA—γ-aminobutyric acid. Adapted from “Gut-Brain-Axis”, by BioRender.com (accessed on 26 August 2022). Retrieved from https://app.biorender.com/biorender-templates.

**Table 1 ijerph-19-11022-t001:** Summary of registered intervention clinical trials of dietary supplementation with probiotics in the prevention or treatment of symptoms or side effects of COVID-19 including mental health issues outcome measures (via https://www.clinicaltrials.gov/, accessed on 25 July 2022).

Clinical Trials Identifier and Status	Location	Official Title	Evaluation	Hypothesis	Intervention	Mental Health Outcomes Measures	Participants	Reference
**NCT04507867**, Completed	Toluca de Lerdo, Mexico	Effect of an NSS to reduce complications in patients with COVID-19 and comorbidities in stage III	Randomized clinical trial	The nutritional support system will reduce the complications of patients with COVID-19 in stage III with comorbidities	Nutritional support system (NSS), which consists of probiotics *Saccharomyces boulardii* 50 million CFU daily for 6 days orally and placebo	PHQ-9 test *	80	https://clinicaltrials.gov/ct2/show/NCT04507867 (accessed on 25 July 2022)
**NCT03851120**, Active, not recruiting	Jakarta, Indonesia	Promotion of maternal gut microbiota and psychological stimulation on child cognitive development at 6 months of age	Randomized clinical trial and placebo parallel controlled study, followed by a follow-up study at 2 years old	Maternal probiotic + LC-PUFA supported with government program supplements, healthy eating, and psychosocial stimulation affect fetal brain development and later child brain functions and cognitive development in light of the COVID-19 outbreak	Probiotics and 480 mg DHA, psychosocial stimulation, healthy eating education, and placebo	Mother’s depression test, mother quality of life, child cognitive at 4, 6, 25–28 months of age, child brain function at 4 months of age	314	https://www.clinicaltrials.gov/ct2/show/NCT03851120 (accessed on 25 July 2022)
**NCT04884776**, Recruiting	Hong Kong, Hong Kong	Modulation of gut microbiota to enhance health and immunity of vulnerable individuals during COVID-19 pandemic	A double-blinded, randomized, active-placebo controlled study	Modulating the gut microbiota with a microbiome immunity formula can rebalance the gut microbiota in populations at risk of infection, like, patients with type 2 DM and elderlies and can lower the number of hospitalization and reduce side effects associated with COVID-19 vaccination	Microbiome immunity formula 2 sachets daily for a total of 12 weeks (3 *Bifidobacteria*, 10 billion CFU per sachet) and placebo contains active vitamin	EQ-5D-5L questionnaire **	484	https://clinicaltrials.gov/ct2/show/NCT04884776 (accessed on 25 July 2022)
**NCT04950803**, Recruiting	Hong Kong, Hong Kong	A randomized controlled trial of an oral microbiome immunity formula in reducing the development of long-term co-morbidities in recovered COVID-19 patients	Multi-center, triple-blind, randomized, placebo-controlled clinical trial	The oral microbiome immunity formula (SIM01) modulates gut microbiota, enhancing immunity and reducing long-term complications and comorbidities in patients who have recovered from COVID-19	Microbiome immunity formula contains probiotics blend (3 *Bifidobacteria*, 10 billion CFU per sachet) daily for 3 months and a placebo	Neurological system issues	280	https://clinicaltrials.gov/ct2/show/NCT04950803 (accessed on 25 July 2022)
**NCT04922918**, Recruiting	Moralzarzal, Spain	Administration of *Ligilactobacillus salivarius* MP101 in an elderly nursing home during the COVID pandemics	Open-label study	*Ligilactobacillus salivarius* MP101 influences the functional, cognitive, and nutritional status, and the nasal and fecal inflammatory profiles of elderly living in a nursing home highly affected by COVID-19	Administration of *Ligilactobacillus salivarius* MP101 (>9 log10 CFU, daily) for 4 months	GDS/FAST system (cognitive status) ***	25	https://clinicaltrials.gov/ct2/show/NCT04922918 (accessed on 25 July 2022)

* PHQ-9 test evaluates the presence of depressive symptoms based on the criteria of the Diagnostic and Statistical Manual of Mental Disorders version 4; ** EQ-5D-5L questionnaire comprises five dimensions: mobility, self-care, usual activities, pain/discomfort, and anxiety/depression; *** GDS/FAST which is the Global Deterioration Scale/Functional Assessment Staging system, assesses of disease severity in progressive dementing illness.

## Data Availability

Not applicable.

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
