# Peer review of "The Role of Psychobiotics to Ensure Mental Health during the COVID-19 Pandemic—A Current State of Knowledge"

_ijerph, 2022, doi:10.3390/ijerph191711022_

Round 1

Reviewer 1 Report

That review show the role of psychobiotic on mental health and  mental health during COVID-19 Pandemic.  I read this review with great interest. Authors show 4 possibility of the action of psychobiotics on the gut-brain axis and next they presented the role of psychobiotic on the mental health general and mental health related to Covid-19 Pandemic. I wonder what role psychobiotics may play in the context of other viral diseases, e.g. influenza- but of coures is not the subject of your review. The issues have been analyzed by a multifaceted approach. I think that this review may be inspiring to other researchers for creating further research on the effects of psychobiotics- not only mental health.

Author Response

The Autors would like to thank the Reviewer for careful and thorough reading of this manuscript and for the thoughtful comments and constructive suggestions, which help to improve the quality of this manuscript. Our responses to each comment have been marked in blue color below. All the changes in manuscript were marked in colors.

Reviewer 2 Report

The authors reviewed discussion of the potential role of psychobiotic in light of the current problems assessing probiotics therapeutic administration in the prevention or treatment symptoms or side effects of COVID-19. The subject matter of this work is laudable, and the paper is well written. The reviewer thinks, however, that there are a few improvements that should be made.

Major point:

1.       The author should also describe issues of sleep disturbance.

Minor points:

1.       In the Figure 1, the abbreviations should be explained in the caption.

2.       Line 116 and 208, “central nervous system” can be abbreviated as “CNS”, respectively.

3.       Line 217, GIT should be spelled out.

4.       Line 357-361, the font size should be adjusted.

5.       Line 414, “γ-aminobutyric acid ( )” is not needed. Only “GABA” can be used.

Author Response

(The authors gave the same response as above.)

Reviewer 3 Report

The review "The Role of Psychobiotic to Ensure Mental Health During 2 the COVID-19 Pandemic – A Current State of Knowledge" is an interesting work as COVID-19 health crisis must be covered in all aspects. Pointing out mental health issues related to Covid-19 pandemia is relevant, but is more relevant to study potential treatments as proposed in this review with psycobiotics. 

The manuscript is well written and organized, however, I may suggest that the references are correctly associated with the text. REference 4  in line 162, please look for the original publication, do not cite another review for a specific result. Also, reference 26 is not suitable for line 116. Please, review the paragraph from lines 156 to 162. 

Finally, review typos.

Best regards,

Author Response

(The authors gave the same response as above.)

Reviewer 4 Report

Dear Authors,

I have carefully read and revised Review article „The Role of Psychobiotic to Ensure Mental Health During 2 the COVID-19 Pandemic – A Current State of Knowledge“ by Zielinska et al. The article is very actual presenting overall current knowledge of the subject and more prominent hypothesis supported with adequate references. Following well described scientific-grounded  connection between some kind of psychobiotc and specific aspects of mental-wellbeing, the conclusions are carefully made. Thus, the authors had in mind relatively short period since COVID-19 brake out, as well as consequently still present lack of additional clinical trials on psychobiotic  and their efficacy in  treating mental health disorders influenced by COVID-19 pandemic. The article is very well organized and illustrated, I was reading it with interest and pleasure.

I am glad to recommend the Review article „The Role of Psychobiotic to Ensure Mental Health During 2 the COVID-19 Pandemic – A Current State of Knowledge“ by Zielinska et al. to be published in International Journal of Environmental Research and Public Health, in the present form.

Author Response

(The authors gave the same response as above.)

Reviewer 5 Report

Dear Authors,

The manuscript entitled "The Role of Psychobiotic to Ensure Mental Health During the COVID-19 Pandemic – A current State of Knowledge” is an interesting and pertinent topic about the development of new products with an impact on physical and mental health. However, the manuscript is confusing, and it is important to clarify some questions.

I suggest some changes to better understand the article.

Q1.

The manuscript should have a short introduction where the aim of the study is presented and then the literature review...

Q2.

An article about psychobiotics involves a more detailed introduction to these products. Concept, various opinions…

There will be consensus in the literature on the topic of "psychobiotics" should be explored in this article

I suggest reading more articles in this area, namely the article:

Cryan, J. F.; O'Riordan, K. J.; Cowan, C. S.; Sandhu, K. V.; Bastiaanssen, T. F.; Boehme, M.; Codagnone, M. G.; Cussotto, S.; Fulling, C.; Golubeva, A. V., The microbiota-gut-brain axis. Physiological reviews 2019.

Casertano, M.; Fogliano, V.; Ercolini, D., Psychobiotics, gut microbiota and fermented foods can help preserving mental health. Food Research International 2022, 152, 110892.

Q3.

Figure 1 has a very large format, it should be reduced and included on the previous page. It should be included right after your citation.

The legend should be changed. The schema is extensive and does not only refer to psychobiotics, because these gut-brain pathways are important for the mechanism of action of probiotics, prebiotics, and symbiotics, and not simply for psychobiotics.

Q4.

When an abbreviation is defined it should always be used, and only abbreviations that will be used throughout the text should be defined.

Line 340, there is no point in repeating the abbreviation again.

Line 340, “during the coronavirus disease 2019 (COVID-19)”, change to “during the COVID-19”

Line 351, “pre-COVID psychotropic”, include full information “pre-COVID-19…”

Line 357-360, the font size does not match the formatting of the text. Please, rectify!

acceptance model”

Line 366 – Rectify “EQ-5D-5L” – “**EQ-5D-5L”

Line 367 – Include “***GDS/FAZ”

Line 379 - the reference to the mentioned study is missing.

Kato and collaborators’ (2016) study refers to a study with probiotics and not psychobiotics, it is important to understand the distinction between these classifications

Line 378-379 - “Several studies have suggested that the administration of psychobiotics can be effective in stress management.” but then only studies with probiotics are presented…

Line 394-396 “Akkasheh et al. study was designed to determine the effects of probiotic intake on symptoms of depression and metabolic status in patients with MDD (major depressive disorder).”

Article referencing missing again!!!

Line 396 – “MDD (major depressive disorder)” change to “Major Depressive Disorder”

First the expression and then the abbreviation!!

Abbreviations should only be defined if they are used several times. In this case there is no point in defining the abbreviation because it will not be used at all…

Line 397-398 - “beneficial effects on Beck Depression Inventory total scores (P=0.001) compared with the placebo… “

Clarify the sentence, what are the beneficial effects?

Line 398-399 – “In the study of Steenbergen et al. “ change to “In the study of Steenbergen et al. (2015)“ or “In the study of Steenbergen et al. ….. [91]“

The correct reference of the study is missing again!!!

There are too many references, some of which do not make sense, there should be a selection of the references used.

The article is interesting but confusing. It is important to clarify well the concepts of probiotics and psychobiotics...

Author Response

(The authors gave the same response as above.)
